# Origin of the Fastest 5 km, 10 km and 25 km Open-Water Swimmers—An Analysis from 20 Years and 9819 Swimmers

**DOI:** 10.3390/ijerph182111369

**Published:** 2021-10-29

**Authors:** Aldo Seffrin, Beat Knechtle, Rodrigo Luiz Vancini, Douglas de Assis Teles Santos, Claudio Andre Barbosa de Lira, Lee Hill, Thomas Rosemann, Marilia Santos Andrade

**Affiliations:** 1Department of Physiology, Federal University of São Paulo, São Paulo 04021-001, Brazil; netoseffrin@gmail.com (A.S.); marilia1707@gmail.com (M.S.A.); 2Medbase St. Gallen Am Vadianplatz, 9000 St. Gallen, Switzerland; 3Institute of Primary Care, University of Zurich, 8006 Zurich, Switzerland; thomas.rosemann@usz.ch; 4Center for Physical Education and Sports, Federal University of Espírito Santo, Vitória 29075-910, Brazil; rodrigo.luiz.vancini@gmail.com; 5Faculty of Physical Education, State University of Bahia, Teixeira de Freitas 45992-255, Brazil; datsantos@uneb.br; 6Human and Exercise Physiology Division, Faculty of Physical Education and Dance, Federal University of Goiás, Goiânia 74690-900, Brazil; andre.claudio@gmail.com; 7Department of Pediatrics, McMaster University, Hamilton, ON L8S 4L8, Canada; hilll14@mcmaster.ca

**Keywords:** water sport, endurance, origin, nationality

## Abstract

In elite pool swimmers competing at world class level, mainly athletes from the United States of America and Australia are dominating. Little is known, however, for the nationality of dominating swimmers in elite open-water long-distance swimming races such as the official FINA races over 5 km, 10 km and 25 km—held since 2000. The aim of this study was to investigate the participation and performance trends by nationality of these elite open-water swimmers. Race results from all female and male swimmers competing in 5 km, 10 km and 25 km FINA races between 2000 and 2020 were analyzed. A total of 9819 swimmers competed between 2000 and 2020 in these races. The five countries that figure most times among the top ten in 5 km, 10 km and 25 km races over the years were Italy, Germany, Russia, Brazil and the Netherlands. In 10 km races, considering the all the athletes from each country, male athletes from Germany, Italy, and France presented faster race times than the other countries. In 10 km, female athletes presented no significant difference among the countries. In 5 and 25 km races, there were no differences between countries, for male and female athletes. Moreover, comparing only the 10 best results (top 10) from each country, there were no differences between countries in 5 km, 10 km and 25 km, for male and female athletes. Men were faster than women for all three distances. In summary, male swimmers from Europe (i.e., Germany, Italy, France) are dominating the 10 km FINA races. In the 5 km and 25 km FINA races, there is no dominating nationality, but among the top five countries in the top 10 over the years, three are European countries.

## 1. Introduction

Competitive open-water swimming (OWS) is a relatively young sports discipline to the Olympic program, but has been a feature of the Fédération Internationale de Natation (FINA) World Championships since 1991 [1]. As the popularity of OWS increased over the years, the 10 km marathon swim race made its debut at the 2008 Olympic Games in Beijing [2]. Although the current Olympic program only offers the 10 km event, the FINA World Championships Ligue Européenne de Natation (LEN) European Championships have an expanded programs including the 5 km, 10 km and 25 km distances, and a recently introduced mixed team relay event [3,4].

The increased number of competitions have stimulated the interest in OWS worldwide and as a result, the number of participants has substantially increased in the last years [5]. Secondly, it is interesting to note that there has been an exponential growth in women participating [5], including a trend towards an over overall higher competitive level of women than of men [5,6,7]. It has been shown that women were faster than men in the ‘Triple Crown of Open Water Swimming’ with ‘Catalina Channel Swim’, ‘English Channel Swim’ and ‘Manhattan Island Marathon Swim’ [8]. However, in the 5 km, 10 km and 25 km FINA races, men were faster than women in all race distances [9,10] although women improved their performance in the 10 km race distance, but not for the other distances [10,11]. Interestingly, in the 3000 m FINA World Championships between 1992 and 2014, women were not able to reduce the sex difference in performance to men across years [12].

Regarding to the origin of elite pool swimmers at world class level [13,14], mainly athletes from the United States of America (USA) and Australia (AUS) were more likely to finish in the Top 10 or medal positions [15,16]. However, regarding the relationship between the origin of open-water long-distance swimmers and their performance little is known, and the few published data are not from World Championships [8,17,18,19]. In the ‘English Channel Crossing’ between 1875 and 2013, most swimmers were from Great Britain (GB), the USA, AUS and Ireland. Secondly, the fastest swim times were achieved by athletes from the USA, AUS and GB [18] and in the ‘Strait of Gibraltar’, local Spanish swimmers were the fastest [19]. In long-distance open-water events in the ‘Triple Crown of Open Water Swimming’ from 1875 to 2017 (‘Catalina Channel Swim’, ‘English Channel Swim’ and ‘Manhattan Island Marathon Swim’) the fastest swimmers were from AUS, USA, GB and Canada [8] were recorded the fastest performances. Therefore, there is little knowledge from where the swimmers competing in world championships originate from [9].

Therefore, the aim of the study was to investigate the participation and performance trends of elite open-water swimmers competing 5 km, 10 km and 25 km races held since 2000. Based upon existing findings for pool and open-water swimmers we hypothesized that the fastest swimmers competing in that discipline would also originate from the USA and AUS.

## 2. Methods

### 2.1. Ethical Approval

This study was approved by the Institutional Review Board of Kanton St. Gallen, Switzerland, with a waiver of the requirement for informed consent of the participant as the study involved the analysis of publicly available data (EKSG 01-06-2010).

### 2.2. Data

Race results from all female and male swimmers competing in 5 km, 10 km and 25 km FINA races between 2000 and 2020 were obtained from the FINA website [4]. From the entire FINA website database of the competitions of 5, 10 and 25 km races, the following data were selected to tabulation and subsequent analysis: nationality, sex, race time, race distance (5, 10 or 25 km), competition date and local.

### 2.3. Statistical Analysis

Descriptive data were presented by mean, standard deviation, maximum and minimum values. In order to compare race time between sexes, Student-*t* test for independent samples was used. The mean values of the entire sample and top 10 results of each country and sex were selected for analysis. Data did not follow a normal distribution nor had homogeneous variances according to Shapiro–Wilk and Levene’s test, respectively. A generalized linear model (GLM) with a gamma or tweedie probability distribution and identity or log link function was used to assess the effect of nationality of the athlete and the advancement of the years on race time for the entire sample and for top 10 sample, the method of choosing the distribution of the dependent variable and the link function was the Akaike information criterion (AIC), using its lowest value. For this analysis, the nationalities were grouped into six groups, the five nationalities that were most present in the top 10 times in each event and one group with the other nationalities by sex. Differences found were investigated with posthoc Bonferroni test. The level of significance was set at 0.05. SPSS version 26.0 (SPSS, Inc., Chicago, IL, USA) was used for all statistical analyses.

## 3. Results

In order to compare the performance of the countries that participated in the 5, 10 or 25 km swim events, the 5 countries that were most often among the top 10 were first selected (Table 1). The other countries were grouped into a single group called “Others” (Table 1). First, the mean value of each swim distance of all participants from each country (divided by sex) was compared. After this initial analysis, the mean values of each swim distance of the top 10 swimmers of each country were compared. The results section was divided in these two parts. First, data from the entire sample are presented, and, secondly, data from the top 10 comparison are presented.

### 3.1. Comparison among All Athletes from Each Country

A total of 9819 swimmers competed between 2000 and 2020 in 5, 10 and 25 km races. Most of the swimmers (76.1%) competed in the 10 km (*n* = 7476, 3227 women and 4249 men), followed by 5 km (16%, *n* = 1575, 740 women and 835 men) and 25 km (7.9%, *n* = 768, 323 women and 445 men).

The 5 km event (*n* = 1575) had a total mean time of 01:00:15 ± 05:12 (minimum 00:51:17/maximum 01:47:37) h:min:s, the 10 km event (*n* = 7476) had a mean time of 02:02:21 ± 11:43 (minimum 01:29:51/maximum 03:01:00) h:min:s, while the 25 km (*n* = 768) had a mean time of 05:23:07 ± 27:30 (minimum 04:10:41/maximum 07:02:38) h:min:s for the entire sample. The time spend in each distance by each sex were presented in Table 2.

Regarding nationality, there were observed no significant differences on the mean time in the 5 km events for females [x^2^ (5) = 3.608, *p* = 0.607, AIC = 10,297.59] (Figure 1a) and males (Figure 1b) [x^2^ (5) = 9.706, *p* = 0.084, AIC = 11,669.47]. In addition, there was a significant effect of the advancement of the years on the mean time in the 5 km events for males [x^2^ (1) = 27.880, *p* < 0.001], a reduction of 11,906 s/year [IC = −16.325/−7.486], but not for females [x^2^ (1) = 2.978, *p* = 0.084] (Figure 2a).

For 10 km events, no significant differences on the mean time were found based on the nationality of the athlete for females [x^2^ (5) = 6.035, *p* = 0.303, AIC = 51,047.39] (Figure 1c), but there were significant differences on the mean time in the 10 km events for males [x^2^ (5) = 26.388, *p* < 0.001, AIC = 66,358.82], where Germany, Italy and France presented significantly better times than the other countries (Figure 1d). Besides that, there was a significant effect of the advancement of the years on the mean time in the 10km events for females [x^2^ (1) = 50.672, *p* < 0.001], a reduction of 25.543 s/year [IC = −32.576/−18.510], and for males [x^2^ (1) = 76.638, *p* < 0.001], a reduction of 0.004 s/year [IC = −0. 005/−0.004] (Figure 2b).

No significant differences on the mean time of 25 km events were found based on nationality of the athlete for females [x^2^ (5) = 9.417, *p* = 0.094, AIC = 5617.71] (Figure 1e) and males [x^2^ (5) = 1.384, *p* = 0.926, AIC = 7755.62] (Figure 1f). Moreover, there was a significant effect of the advancement of the years on the mean time in the 25 km events for females [x^2^ (1) = 10.683, *p* = 0.001], a reduction of 65.069 s/year [IC = −104.089/−26.049], and for males [x^2^ (1) = 12.197, *p* < 0.001], a reduction of 60.996 s/year [IC = −95.227/−26.765] (Figure 2c).

### 3.2. Comparison between Top 10 Athletes from Each Country

Comparing the race time mean among the top ten athletes from each nationality, there were no significant differences on the mean time in the 5 km events for females [x^2^ (5) = 3.903, *p* =0.5.64, AIC = 2269.27] (Figure 3a) and males (Figure 3b) [x^2^ (5) = 2.002, *p* = 0.849, AIC = 2264.16] based on the nationality of the athlete. In addition, there were no significant effect of the advancement of the years on the mean time in the 5 km events for females [x^2^ (1) = 0.095, *p* = 0.759] and males [x^2^ (1) = 1.330, *p* = 0.249] (Figure 4a).

For 10 km events, no significant differences on the mean time were found based on the nationality of the athlete for females [x^2^ (5) = 1.936, *p* = 0.858, AIC = 2870.49] (Figure 3c), and males [x^2^ (5) = 6.852, *p* = 0.232, AIC = 2825.39] (Figure 3d). Besides that, there was a small but significant effect of the advancement of the years on the mean time in the 10 km events for females [x^2^ (1) = 29.530, *p* < 0.001], a reduction of 0.006 s/year [IC = −0.009/−0.004], and for males [x^2^ (1) = 34.709, *p* < 0.001], a reduction of 0.004 s/year [IC = −0.010/−0.005] (Figure 4b).

In the same way, no significant differences on the mean time of 25 km events were found based on nationality of the athlete for females [x^2^ (5) = 5.415, *p* = 0.367, AIC = 2933.24] (Figure 3e) and males [x^2^ (5) = 0.795, *p* = 0.977, AIC = 2934.96] (Figure 3f). Moreover, there were no significant effect of the advancement of the years on the mean time in the 25 km events for females [x^2^ (1) = 1.614, *p* = 0.204 and for males [x^2^ (1) = 1.359, *p* < 0.244] (Figure 4c).

## 4. Discussion

This study intended to investigate participation and performance trends by nationality of elite open-water swimmers competing in 5 km, 10 km and 25 km races held between 2000 and 2020. We hypothesized that the fastest swimmers competing in these races would originate from the USA and AUS, similarly to the observations of elite pool swimmers.

The main findings considering the entire sample were, (i) Germany, Italy and France presented significantly better times than the other countries in the 10 km races for males, (ii) there were no differences among the countries in the 10 km races for females, (iii) there were no differences among countries in the 5 km and 25 km races for both sexes, (iv) there were significant decreases among the race times over the last 20 years in 5 km, 10 km and 25 km races for both sexes, except for females in the 5 km event, and (v) male athletes presented better race times in 5 km, 10 km and 25 km races than females. Moreover, considering the top 10 athletes from each country, there were no differences among the countries in the 5 km, 10 km or 25 km races for both males and females.

A first important finding, considering the entire sample, was those male swimmers mainly from Europe were the fastest in 10 km events and the hypothesis that swimmers from the United States of America and Australia would be the fastest could not be confirmed. Regarding female athletes, there were no significant differences in swimming times between nations in the 10 km events, despite athletes from Germany, Italy, Brazil, Russia and USA featured more frequently in the top 10 positions. In the 5 km and 25 km events, there was no significant difference in the race times between athletes from different countries for both male and female athletes. Comparing the average time of the 10 best athletes from each country, there were also no significant differences between countries. Obviously, the density of high-performance athletes seems to be high in the countries where both all swimmers and the top ten per country were the fastest.

Previous studies investigated the aspect of nationality in long-distance open-water swimming events where non-elite swimmers were performing in a solo event such as the ‘English Channel Crossing” [17,18,20], the “Strait of Gibraltar” [19], the “Triple Crown” [8], “Manhattan Island Cross [6], “Maratona del Golfo Capri-Napoli” [7] and the “Robben Island Crossing” [21]. Although some swimmers from the USA and AUS did feature in the solo OWS events could possibly be attributed to non-elite nature of the events. Ultra-long OWS events are often performed on an individual basis by recreational swimmers or swimmers seeking an individual challenge. These events are often self-funded and as a result selects for a very specialized group who wish to complete these challenges, and not necessarily in the fastest time, therefore the characteristics of athletes are very different from those who participate in World Championships.

We found that female swimmers from Italy achieved the best performances and male swimmers from Italy the second-best performances behind Germany although they were not significantly faster than swimmers from other countries. The finding that Italians are among the best in this sport discipline is not accidentally. A very actual study investigating 9247 female and male swimmers competing between 1986 and 2019 in the 3000 m open-water swimming Master World Championships found that female and male swimmers from Italy were the fastest during this period of 27 years and 15 editions all over the world [22].

A potential explanation that swimmers from Italy and Germany are the best in the world in this sports discipline could be their national efforts. For example, Italy has its ‘Circuito Nuoto in Acque Libere’ where open-water swimmers are organized to compete in different formats of open-water swimming [23]. In addition, in Germany, open-water swimming is promoted with information about training, equipment and events [24].

In terms of performance, considering the top 10 results from each country there were no improvement in mean time in the last 20 years, and although a significant decrease in mean times was observed considering the entire sample, sometimes the effect was very small, for example in male 10 km events a reduction of 0.004 s/year was observed. In the 25 km events, there was a slightly greater decrease in time over the years (a reduction of 65.069 s/year for females and a reduction of 60.996 s/year for males).

In other open-water long-distance swimming events, a more expressive improvement in female and male performance has been reported [5,10,13,14,25]. A potential explanation for these small performance improvements over the years could be the relatively short time frame in this specific sports discipline (2000–2020) compared to other open-water long distance swimming events such as the ‘English Channel Crossing’ [17,26]. It is also interesting to note a large variation in finishing times among the years in the 5, 10 and 25 km events, which may be attributable to the differences in sea or weather conditions. For example, if in one year the athletes swim against the current and in the other year they swim with the current, there can be a great impact on the difference in performance between years for both sexes [27].

Moreover, it is important to consider that the FINA OWS events (5 km, 10 km and 25 km) are measured distances with very little variation compared to the ultra-long solo OWS events. Therefore, as the distances are pre-determined for the 5 km, 10 km and 25 km, swimmers and their coaches are able to plan and execute training accordingly in order to complete the races in the shortest time possible. In contract, the ultra-long solo OWS events require swimmers to complete a distance set between two geographic points [28], and as such may be influenced by several environmental factors, thus changing the objective to reflect completing the challenge rather than defeating a field of other competitors [29,30].

Another important finding was that men were faster than women in all three disciplines. Generally, male performance is better than female performance in a variety of sports disciplines [31,32,33,34]. However, in some long-distance open-water swimming events, women were faster than men [20]. A potential explanation that women were slower than men in these 5 km, 10 km and 25 km FINA races could be the fact that these events are not long enough for potential physiological and body composition differences to influence the outcome [7,8]. Previous studies examining ultra-long OWS athletes hypothesized that body composition may have an important role in marathon swimming performance [7,8]. However, FINA races have banned the use of high-performance wetsuits which could potentially mitigate any performance advantage that may be conferred by sex during these events [27]. Moreover, as the events are much shorter than other solo OWS events, which limits water immersion time, which has been reported as a factor affect endurance during marathon swimming [35,36,37].

A limitation of the present study is that the present study did not assess the effect of age in performance [38]. Thus, the results show comparisons of the best times obtained among all athletes, but it is possible that if the differences in performance of each nationality by age group are studied, different results will be found. A further limitation is that environmental aspects such as temperatures were not considered since open-water swimming races can be characterized by extreme environmental conditions (e.g., water temperature, tides, currents, and waves) which might have an impact on performance, influencing both tactics and pacing [39]. In addition, physiological and anthropometric aspects [40] and race strategies [41,42] were not considered. Finally, the race time of the athletes who competed each year were analyzed, but we are not aware of the possibility that there were athletes who competed in more than one competition.

## 5. Conclusions

In summary, male swimmers from Europe (Germany, Italy and France), are dominating the 10 km FINA races, but there are no specific country dominations in 5 km or 25 km events for both sexes. The results do not change when all swimmers and only the top ten per country are considered. Future studies might investigate the aspect of nationality in other swimming disciplines such as in master swimming, elite pool swimming and ice swimming.

## Figures and Tables

**Figure 1 ijerph-18-11369-f001:**
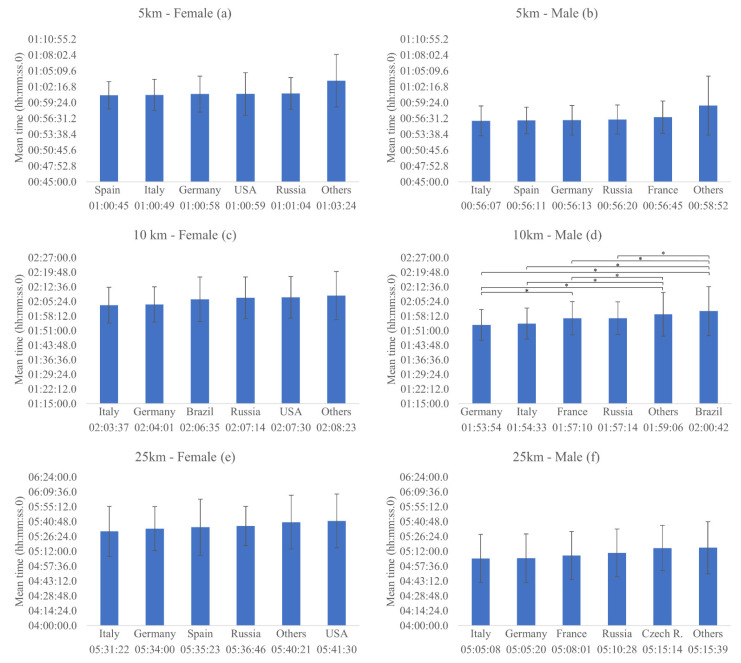
Mean race times and significant differences among nationalities, regarding the entire sample, in the female and male 5, 10 and 25 km events. * *p* < 0.005.

**Figure 2 ijerph-18-11369-f002:**
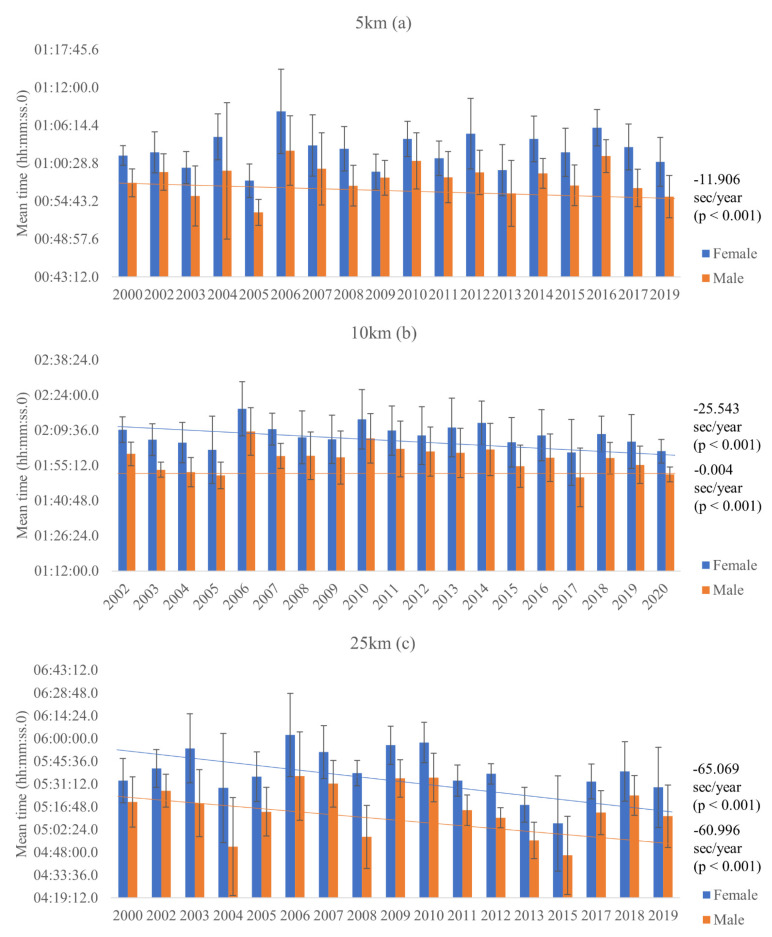
Mean values, of entire sample results, for each race over the years for both sexes in 5, 10 and 25 km events.

**Figure 3 ijerph-18-11369-f003:**
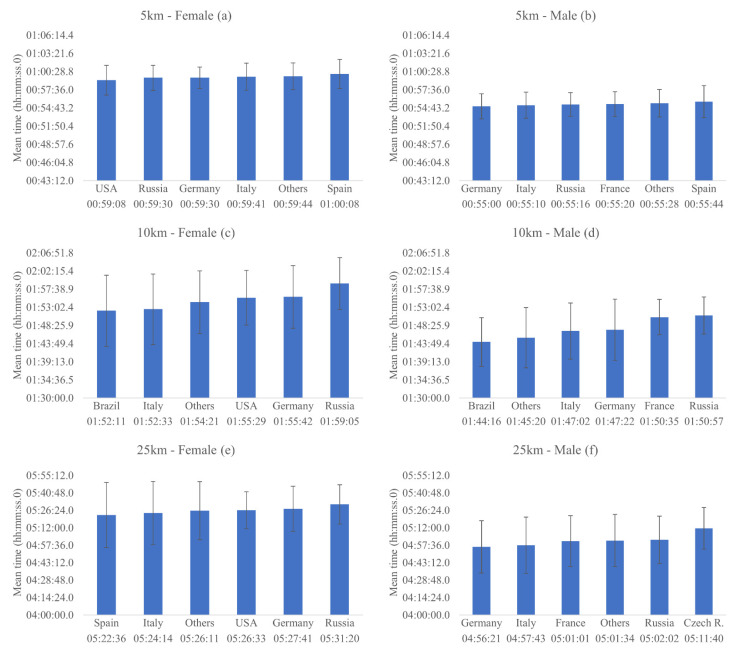
Mean race times for each nationality, regarding the top 10 sample, in the female and male 5, 10 and 25 km events.

**Figure 4 ijerph-18-11369-f004:**
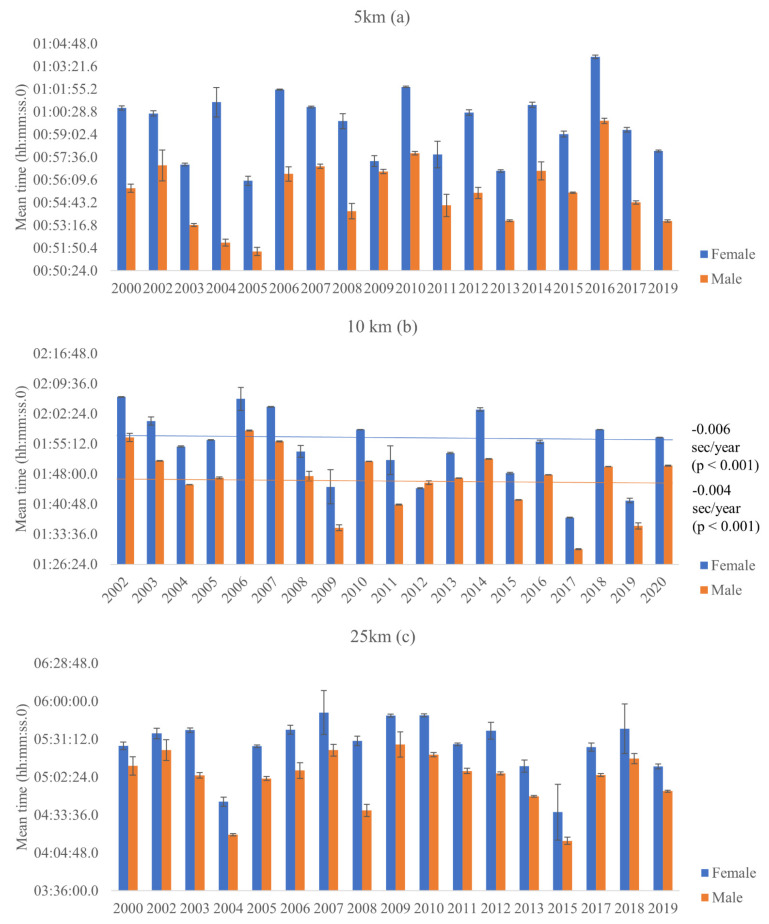
Mean values, of top 10 sample results, for each race over the years for both sexes in 5, 10 and 25 km events.

**Table 1 ijerph-18-11369-t001:** Sum of nationality that figure in the top 10 times in each event by sex.

Distance	Female Nationality	Female (*n*)	Male Nationality	Male (*n*)
5 km	Italy	31	Italy	35
	Germany	19	Germany	26
	Russia	19	Russia	25
	USA	17	France	19
	Spain	12	Spain	14
	Others	82	Others	61
10 km	Germany	29	Germany	33
	Italy	25	Italy	28
	Brazil	22	Russia	17
	Russia	14	France	14
	USA	13	Brazil	13
	Others	87	Others	85
25 km	Russia	28	Italy	30
	Italy	27	France	26
	Germany	21	Russia	24
	Spain	14	Germany	12
	USA	11	Czech R.	9
	Others	68	Others	68

**Table 2 ijerph-18-11369-t002:** Race time in each event by sex.

	Sex				
	Female		Male		
Event	*n* (%)	Time	*n* (%)	Time	*p* Value *
5 km	740 (47)	01:02:40 ± 04:29 (00:55:40/01:31:43)	835 (53)	00:58:07 ± 04:50 (00:51:17/01:47:37)	<0.001
10 km	3227 (43.2)	02:07:29 ± 11:20 (01:37:29/03:01:00)	4249 (56.8)	01:58:27 ± 10:27 (01:29:51/02:46:56)	<0.001
25 km	323 (42.1)	05:37:36 ± 24:35 (04:18:28/07:02:38)	445 (59.1)	05:12:36 ± 24:34 (04:10:41/06:39:33)	<0.001

*n* = sample size; % = percentage values; * *p* < 0.05 (race time for male group was significantly lower than for female group).

## Data Availability

Data were obtained from https://www.fina.org and are available from the authors upon request.

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
