# Peer review of "Origin of the Fastest 5 km, 10 km and 25 km Open-Water Swimmers—An Analysis from 20 Years and 9819 Swimmers"

_ijerph, 2021, doi:10.3390/ijerph182111369_

Round 1
Reviewer 1 Report
I was invited to revise this paper that aimed to assess trends in swimming race performances across countries during a 20-years period.
In my opinion this study does not represent a priority for public health and focused on a topic that does not implies interest in medical professionals.
In addition, Authors tried to analyze data publicly available and do not add particular analyses or experiments in their report.
About the paper, I have some comments:
- Introduction is adequate;
- Methods should be deeply described;
- In statistical analysis Authors reported to perform the AIC to predict the most accurate model, but they did not reported it;
- Authors should analyze each participant only once. Each athlets can change his performance during time so it is a age/time dependent variable;
- About figures, Authors should use boxplots with error/distribution bar instead of column;
- Line 120 - There is an error in the table, without caption;
- Figures 2 and 4 are unreadable. If Authors performed a GLM, please use a different kind of graph;
- Limitation section is too poor.
Author Response
Reviewer #1
I was invited to revise this paper that aimed to assess trends in swimming race performances across countries during a 20-years period.
In my opinion this study does not represent a priority for public health and focused on a topic that does not implies interest in medical professionals.
In addition, Authors tried to analyze data publicly available and do not add particular analyses or experiments in their report.
About the paper, I have some comments:
Introduction is adequate;
Answer: Thank you about your positive comment.
Methods should be deeply described;
Answer: We thank the expert reviewer for his/her comment. Data about the variables selected for further analysis has been included in the method section.
In statistical analysis Authors reported to perform the AIC to predict the most accurate model, but they did not reported it;
Answer: We thank the expert reviewer for his/her comment. The AIC itself is not relevant enough to be presented in the study, but it is important that the reader knows what was done, for this reason we chose to keep this explanation in the methodology section.
Authors should analyze each participant only once. Each athlets can change his performance during time so it is a age/time dependent variable;
Answer: We thank the expert reviewer for his/her comment. Unfortunately, it is not possible to do this type of analysis.
About figures, Authors should use boxplots with error/distribution bar instead of column;
Answer: We thank the expert reviewer for his/her comment. All figures have been reorganized, and we included SD for all the bars representing mean values.
Line 120 - There is an error in the table, without caption;
Answer: Thank you for calling our attention to this point. The mistake has been corrected.
Figures 2 and 4 are unreadable. If Authors performed a GLM, please use a different kind of graph;
Answer: We thank the expert reviewer for his/her comment. All figures have been reorganized
Limitation section is too poor.
Answer: We thank the expert reviewer for his/her comment. We expanded that section as requested
Reviewer 2 Report
This version of the manuscript has improved dramatically and became a very interesting paper.Author Response
Reviewer #2
This version of the manuscript has improved dramatically and became a very interesting paper.
Answer: Thank you about your positive comment.
Reviewer 3 Report
In this study the authors have demonstrated the participation and performance trends of elite open-water swimmers by taking nationality and gender under consideration. The manuscript has been crafted registering data from 9819 swimmers, following them for 20 years of time, which is very impressive. Here are some comments/concerns the authors need to address:
- Figure 1 and 3: The authors are requested to put a Y axis (time) in the bar graphs. The bar graphs are not represented correctly unless there is a time range axis.
- For all the bars representing the mean value, it is recommended to keep SD or SEM error bars.
- While comparing the countries the authors are requested to put a line over the bars to denote which country groups are being compared and put a p value on to top of it.
- The authors have represented comparison between male and female subjects over the years (bar graphs for each year), however, below those figures the authors are requested to put a bar graph which has male vs female comparison collectively for 20 years for each of the 5, 10 and 25 km events. Compare the groups and put a p value.
Author Response
Reviewer #3
In this study the authors have demonstrated the participation and performance trends of elite open-water swimmers by taking nationality and gender under consideration. The manuscript has been crafted registering data from 9819 swimmers, following them for 20 years of time, which is very impressive. Here are some comments/concerns the authors need to address:
Figure 1 and 3: The authors are requested to put a Y axis (time) in the bar graphs. The bar graphs are not represented correctly unless there is a time range axis.
Answer: Thank you for calling our attention to this point. We have reorganized Figure 1 and 3, and Y axis (time) has been included in the bar graphs.
For all the bars representing the mean value, it is recommended to keep SD or SEM error bars.
Answer: We are grateful for the contribution, we included SD for all the bars representing mean values.
While comparing the countries the authors are requested to put a line over the bars to denote which country groups are being compared and put a p value on to top of it.
Answer: On more time, we are grateful for the enriching comment, and a line over the bars has been included. Please, let us know if these changes do not meet with your expectation.
The authors have represented comparison between male and female subjects over the years (bar graphs for each year), however, below those figures the authors are requested to put a bar graph which has male vs female comparison collectively for 20 years for each of the 5, 10 and 25 km events. Compare the groups and put a p value.
Answer: We thank the expert reviewer for his/her comment. Male vs female comparison collectively for 20 years for each event and p values are presented in Table 2. We chose not to include a figure with this information, because it would be repeated. In addition, the table is more complete, as there is also information about the number of athletes of each sex in each event and the percentage values.
Reviewer 4 Report
I am sure that this is of interest to people who like to swim. However, to others it seems rather boring, or lack of concern. The only issue here to me, is your lack of importance. Why is this study so important? What are you trying to show? Bring in your literature some reasons why we need to show the nationalities of these swimmers. Make some conclusions that are appropriate for the social justice and reflection of society of these countries.
Author Response
Reviewer #4
I am sure that this is of interest to people who like to swim. However, to others it seems rather boring, or lack of concern. The only issue here to me, is your lack of importance. Why is this study so important? What are you trying to show? Bring in your literature some reasons why we need to show the nationalities of these swimmers. Make some conclusions that are appropriate for the social justice and reflection of society of these countries.
Answer: We thank the expert reviewer for his/her comment. The knowledge about the origin of the most successful athletes allows future studies to be developed in order to evaluate the specificities of these regions/peoples capable of generating differentiated results in aquatic marathon events. In addition to the origin of the most successful athletes, it is interesting to note that there has been an exponential growth in women’s participant in open water swimming events, including a trend of great improvement in the performance of women in long-distance events. Although women are participating more and improving performance, we do not know if the improvement observed in women is different from the improvement in performance that men are also showing. Therefore, in order to compare the evolution of performance between the sexes over the last 20 years and also in order to compare nationality and verify if there is any country that stands out in terms of performance, we understand that the study is relevant for the area of knowledge in which it belongs.
Round 2
Reviewer 1 Report
Authors did not addressed all point raised.
Author Response
Reviewer #1
I was invited to revise this paper that aimed to assess trends in swimming race performances across countries during a 20-years period.
In my opinion this study does not represent a priority for public health and focused on a topic that does not implies interest in medical professionals.
In addition, Authors tried to analyze data publicly available and do not add particular analyses or experiments in their report.
Answer: The submission was on invitation from an editor of IJERPH.
According to the aims and scope of IJERPH published in the home page of the journal, it is a is a peer-reviewed scientific journal that publishes original articles in the interdisciplinary area of environmental health sciences and public health, and as a comprehensive multi-disciplinary journal, IJERPH is comprised of nineteen major sections including Exercise and Health. Therefore, we believe that our manuscript is very interesting for this area of knowledge and for medical professionals and other professionals who work with sport and exercise science. It is noteworthy that sports medicine is a new area within medicine that is very promising, given the importance of sport for the prevention and treatment of various diseases, especially non-communicable ones. In this sense, master sport, the subject addressed in the article, is of special interest, since stimuli for the world population to remain active is of fundamental interest to health and world economy.
Despite the data were publicly available, they were not organized to answer the manuscript aim, which was to investigate the participation and performance trends of elite open-water swimmers competing 5 km, 10 km and 25 km races held since 2000. An extensive statistical analysis has been performed through generalized linear model (GLM) with a gamma (5km and 10km events) or tweedie (25km events) probability distribution and identity or log (only for 10km/males) link function to assess the effect of nationality of the athlete and of the advancement of the years on race time for the entire sample. Without this analysis, it would be not possible to conclude about the effect of the nationality or the effect of the advancement of the years on the race time.
About the paper, I have some comments:
Introduction is adequate;
Methods should be deeply described;
Answer: Data about the variables selected for further analysis had already been included in the method section in the previous answer. Please let us know if the changes do not resolve your doubts in this matter.
In statistical analysis Authors reported to perform the AIC to predict the most accurate model, but they did not reported it;
Answer: The AIC values have been included in the results section.
Authors should analyze each participant only once. Each athletes can change his performance during time so it is a age/time dependent variable;
Answer: The lack of information about the presence of athletes who participated in more than one competition was included in the study limitations.
About figures, Authors should use boxplots with error/distribution bar instead of column;
Answer: All figures had already been reorganized in the last manuscript version.
Line 120 - There is an error in the table, without caption;
Answer: Thank you for calling our attention to this point. The mistake has been corrected.
Figures 2 and 4 are unreadable. If Authors performed a GLM, please use a different kind of graph;
Answer: All figures had already been reorganized in the last manuscript version. Please let us know if the changes do not resolve your doubts in this matter.
Limitation section is too poor.
Answer: We expanded that limitation section even further, as you requested.
This manuscript is a resubmission of an earlier submission. The following is a list of the peer review reports and author responses from that submission.
Round 1
Reviewer 1 Report
I was invited to revise the paper entitled "Origin of the fastest 5 km, 10 km and 25 km open-water swimmers". It aimed to investigate the participation and performance trends of elite open-water swimmers competing 5 km, 10 km and 25 km races held since 69 2000.
I do not understand why Authors submitted their research in IJERPH. In my opinion the subject of this paper does not match with the journal scope.
About the research, I have some comments:
- Authors should report age as covariate. It is a foundamental variable to be included.
- Tables 1 and 3 should be reorganized: in table 1 gender should be reported as column and Authors should add a p-value column, About table 3, a nationality column should be removed.
- Authors reported to perfom a GLM but presented result in Fig 3 as a Jointpoint regression analysis. It is unclear.
- Discussion are poor and should be improved.
In conclusion, it is unclear what this paper adds to the knowledge and how it can be important for public health.
Author Response
Reviewer 1
I was invited to revise the paper entitled "Origin of the fastest 5 km, 10 km and 25 km open-water swimmers". It aimed to investigate the participation and performance trends of elite open-water swimmers competing 5 km, 10 km and 25 km races held since 69 2000.
I do not understand why Authors submitted their research in IJERPH. In my opinion the subject of this paper does not match with the journal scope.
About the research, I have some comments:
Authors should report age as covariate. It is a fundamental variable to be included.
Answer: We are grateful for the reviewer's contribution. The aim of the present study was to study the nationality of the fastest swimmers in the world in 5 km, 10 km and 25 km open water swimming events. Using age as a covariate, could favor older but slower athletes, which would be a complementary analysis of the present study. In the same way, to study the nationality of the fastest of each age group also would be a complementary topic of interest, not addressed in this study. However, age is not available in database records so this other aim could be answered. The lack of information about nationality differences in each age group was added as one of the study limitations.
Tables 1 and 3 should be reorganized: in table 1 gender should be reported as column and Authors should add a p-value column, about table 3, a nationality column should be removed.
Answer: We are grateful for the enriching comment, we have reorganized Table 1. The p values for comparison between sexes has been added, as requested by you. Percentage values of male and female swimmers were also included. Regarding table 3, we would like to emphasize that this is a descriptive table with the aim of illustrating which nationalities figures in top 10 in each distance and, therefore, were considered for statistical analysis. Note that the first column of nationality refers to female nationality and the second column refers to male nationality. This information has been included in the table to facilitate the reader's understanding. Removing the nationality column would make it useless, so we chose to keep it. We would be grateful to review this position if we have not understood the reviewer's position.
Authors reported to perform a GLM but presented result in Fig 3 as a Jointpoint regression analysis. It is unclear.
Answer: Thank you about your constructive comments. The line graph chosen was not the best to represent the results obtained by the GLM, implying that other analyzes could have been done. We changed it to a bar chart to improve the presentation of the average times accompanied by the downward trend when there was a significant effect of the advancement of the years.
Discussion is poor and should be improved.
Answer: We added the very recent finding for Italian open-water swimmers showing that Europeans are among the best in long-distance open-water swimming. We also added the aspect of promoting open-water swimming in Germany and Italy, the two European countries with the fastest performances.
In conclusion, it is unclear what this paper adds to the knowledge and how it can be important for public health.
Answer: IJERPH covers – regarding the title – environmental research AND public health. See for example a very recent study in this journal Int. J. Environ. Res. Public Health 2021, 18(14), 7606; https://doi.org/10.3390/ijerph18147606. Moreover, you can see other papers published in IJERPH different sections, including Exercise and Health.
Reviewer 2 Report
The paper Origin of the fastest 5km, 10km, and 25 km open-water swimmers, an analysis from 20 years and 10,000 swimmers analyzed data from 9,819 swimmers in open-water competitions in 20 years and reported the nationality, trend of speed of the top 10 positions in all the competitions. There are some interesting points, for example, the comparison between the woman’s speed with man’s speed. However, the paper focused on the top 10 times, and ignored the rest of the swimmers without a rational explanation why the top 10 positions are more important than the rest of the swimmers. And if there is nothing related with the rest of the swimmers, there is no reason to mention 10,000 swimmers in the title.
Figures 1 and 2 show the average time among those players in top 10 positions in individual countries. What do these figures mean? What can I get from these numbers? Why don't the authors show us the top speed of each country, instead of the mean time?
The trend in figure 3 showed significant correlation between man and woman score. Whenever man had a better score, woman had a similar better score. Is there possible systematic variation among all these competitions in different years? It makes the trend in figure 3 very likely created by these variations.
Each competition should have different rules to select the players to attend the competition. And the rule will dramatically shape the competition and directly affect the time of top 10 positions and nationality of players in that competition.
Author Response
Reviewer 2
The paper Origin of the fastest 5km, 10km, and 25 km open-water swimmers, an analysis from 20 years and 10,000 swimmers analyzed data from 9,819 swimmers in open-water competitions in 20 years and reported the nationality, trend of speed of the top 10 positions in all the competitions. There are some interesting points, for example, the comparison between the woman’s speed with man’s speed. However, the paper focused on the top 10 times, and ignored the rest of the swimmers without a rational explanation why the top 10 positions are more important than the rest of the swimmers. And if there is nothing related with the rest of the swimmers, there is no reason to mention 10,000 swimmers in the title.
Answer: We appreciate the reviewer's contribution, although the study population consisted of 9,819 swimmers, 1,078 were selected on the basis of performance in order to compare dominant nationalities in elite open water long-distance swimming runs. Thus, we chose to readjust the title to better describe the analysis performed. We could perform an extra analysis or restructure the model to cover the entire population, however, an extra analysis would be an element of confusion and readjusting the model would require an overall change in the scope of work. We believe that readjusting the title is an adequate solution. As the number of participants from each country varies, using the entire sample can generate a bias in interpreting the results. Countries with a large participation of athletes have higher average values of the race time, because there are many slow athletes, despite have some fast athletes too.
Figures 1 and 2 show the average time among those players in top 10 positions in individual countries. What do these figures mean? What can I get from these numbers? Why don't the authors show us the top speed of each country, instead of the mean time?
Answer: We appreciate the reviewer's position, which gives us the opportunity to justify presenting the average time based on our practical experience, since the average time obtained by athletes is the performance index most used by athletes and coaches to monitor and compare performance. In addition, frequently literature data also present the swimming performance with the average values of the results obtained in the competition (Fischer et al., 2013; Seffrin et al., 2021). Thus, figures 1 and 2 show the average time obtained by the best athletes from the nationalities most present in the top 10. In addition to serving as a parameter for athletes and coaches, it makes it possible to compare the performance between the dominant nationalities.
Fischer, G., Knechtle, B., Rüst, C. A., and Rosemann, T. (2013). Male swimmers cross the English Channel faster than female swimmers. Scandinavian Journal of Medicine and Science in Sports 23. doi:10.1111/SMS.12008.
Seffrin, A., DE Lira, C. A., Nikolaidis, P. T., Knechtle, B., and Andrade, M. S. (2021). Age-related performance determinants of young swimmers in 100- and 400-m events. The Journal of sports medicine and physical fitness. doi:10.23736/S0022-4707.21.12045-6.
The trend in figure 3 showed significant correlation between man and woman score. Whenever man had a better score, woman had a similar better score. Is there possible systematic variation among all these competitions in different years? It makes the trend in figure 3 very likely created by these variations.
Answer: We would like to thank the reviewer for the constructive comment. It is interesting to note a large variation, which is similar between sexes, in finishing times among the years in 5, 10 and 25 km events, which may be attributable to the differences in sea or weather conditions. For example, if in one year the athletes swim against the current and in the other year they swim with the current, there can be a great impact on the difference in performance between years for both sexes. This paragraph has been included in the discussion section.
Each competition should have different rules to select the players to attend the competition. And the rule will dramatically shape the competition and directly affect the time of top 10 positions and nationality of players in that competition.
Answer: We are astonished about the comment that each competition should have different rules to select the players. We are writing a study about swimmers, and we hope that this reviewer has reviewed the manuscript by himself and not delegated to somebody else. The reviewer used the term ‘player’ two times in his comment. In swimming, different conditions can occur, we add this aspect in the limitations.
Round 2
Reviewer 1 Report
I was invited to review the revised version of the paper entitled "Origin of the fastest 5 km, 10 km and 25 km open-water swimmers".
The paper represent only a descriptive analysis of results obtained from swim athletes. The paper does not meet a high standard to be published in this journal. Revisions submitted did not improved the paper.
Reviewer 2 Report
The authors addressed most of my questions. The last question I raised is about the policy of different games. For example, the Boston Marathon's minimum speed is much higher than the local Marathons'. Therefore, the average score in the Boston Marathon is much higher than the local Marathon. Is that the case in open water swimming?
By the way, if the authors can add an error bar on those bar charts, the bar-charts will be much better.